# Women's experiences of maternal and newborn health care services and support systems in Buikwe District, Uganda: A qualitative study

**Marte Bodil Roed**[1]*, **Ingunn Marie Stadskleiv Engebretsen**[1], **Robert Mangeni**[2], **Irene Namata**[3]

1 Department of Global Public Health and Primary Care, Centre for International Health, University of Bergen, Bergen, Norway, 2 Department of Social Work and Social Administration, College of Humanities and Social Sciences, Makerere University, Kampala, Uganda, 3 Faculty of Social Culture and Development Studies, Muteesa 1 Royal University, Masaka, Uganda

* mbodilro@online.no

**Data Availability Statement:** Data cannot be shared publicly because of identifiable sensitive information revealed by participants during interviews and focus group discussions. Data in

## Abstract

### Background

Uganda continues to have a high neonatal mortality rate, with 20 deaths per 1000 live births reported in 2018. A measure to reverse this trend is to fully implement the Uganda Clinical Guidelines on care for mothers and newborns during pregnancy, delivery and the postnatal period. This study aimed to describe women's experiences of maternal and newborn health care services and support systems, focusing on antenatal care, delivery and the postnatal period.

### Methods

We used triangulation of qualitative methods including participant observations, semi-structured interviews with key informants and focus group discussions with mothers. Audio-recorded data were transcribed word by word in the local language and translated into English. All collected data material were stored using two-level password protection or stored in a locked cabinet. Malterud's Systematic text condensation was used for analysis, and NVivo software was used to structure the data.

### Findings

Antenatal care was valued by mothers although not always accessible due to transport cost and distance. Mothers relied on professional health workers and traditional birth attendants for basic maternal services but expressed general discontentment with spousal support in maternal issues. Financial dependency, gender disparities, and lack of autonomy in decision making on maternal issues, prohibited women from receiving optimal help and support. Postnatal follow-ups were found unsatisfactory, with no scheduled follow-ups from professional health workers during the first six weeks.

form of interview transcripts are available from the Centre for International Health, University of Bergen (contact via post@cih.uib.no), upon specific and reasonable request for researchers who meet the criteria for access to confidential data.

**Funding:** The authors received no specific funding for this work.

**Competing interests:** The authors have declared that no competing interests exist.

## Conclusions

Further focus on gender equity, involving women's right to own decision making in maternity issues, higher recognition of male involvement in maternity care and improved postnatal fol-low-ups are suggestions to policy makers for improved maternal care and newborn health in Buikwe District, Uganda.

## Introduction

Uganda continues to have a high neonatal mortality rate, with 20 deaths per 1000 live births reported in 2018. The country is projected to miss the Sustainable Development Goal (SDG) 3.2 of a neonatal mortality rate of 12/1000 live births in 2030, if the current trend continues [1]. Contributing factors to the high number of neonatal deaths in Uganda are the country's challenges to implementing multi-level policy strategies for enhancing newborn survival. Policy strategies include implementation of The Uganda Clinical Guidelines (UCG) from 2016 with updates, which have detailed descriptions of recommended health care and medical advice in all aspects involving maternal care [2]. Additional obstacles for optimal maternal and newborn health care include informal out-of-pocket payments for health care services, along with shortage of accessible staff, and low quality of care [3]. WHO defines quality of care as:

> *The extent to which health care services provided to individuals and patient populations improve desired health outcomes. In order to achieve this, health care must be safe, effective, timely, efficient, equitable and people-centred* [4].

Quality of care is contextualized within the phenomena of maternal and newborn support systems. It implies that the mothers have someone to trust and rely on during pregnancy, time of childbirth and the following postnatal period, enhancing the chance for an optimal start in life for the newborns [5]. So that in times of need, or if encountering emergencies, the mothers have a security net around them which can provide practical help and emotional support, such as encouraging mothers to seek professional health care [6]. Evidence from other settings show that women with supportive environments and reliable partners have easier births, healthier babies and better postnatal experiences [7].

The term *postnatal* is defined by WHO as "*the time after birth and up to six weeks (42 days)*". The most vulnerable period for a newborn and its mother is the first month after birth, where the first 24 hours are the most critical. Despite this fact, attention and care for the mother and newborn are often neglected after delivery [8]. The national estimate of mothers receiving postnatal care within two days is only 33%, and studies from Uganda show that giving birth in health facilities is the predominant factor for receiving early postnatal care [9, 10].

Assurance of quality of care through the conjugated components of maternal care is important in ensuring optimal outcome for both the mother and her infant. Family and spousal support through each of these stages are paramount for the mothers' wellbeing [6]. This study aimed to describe women's experiences of maternal and newborn health care services and support systems in Buikwe District, focusing on antenatal care, delivery and the postnatal period.

## Methods

### Study design

In an effort to get a deeper and more elaborated understanding of the topic, we used qualitative methods where data was collected through focus group discussions, key informant interviews,

and participant observations [11]. The findings presented in this paper are part of a broader study named "Health care and support systems for mothers and newborns around delivery in Buikwe district, Uganda" and the methods have been described earlier [12]. The research was part of the fulfilment of a M.Phil. degree in Global Health at the Centre for International Health, University of Bergen.

## Setting and population

We conducted research in 7 villages in Buikwe district, in east central Uganda. Buikwe has a total population of approximately 423,000 (2014), where the majority of the people relies on subsistence farming [13]. Most people travel on foot, or they use bicycles or scooters for transportation. Some villages are remote from the closest health facility and traveling at night can be hazardous on narrow dirt roads. Nyenga town contains Saint Francis Hospital and Saint Francis School of Nursing and Midwifery. Kabizzi village includes a Health Centre III for the public in the area, which provide both in-and out-patient services, including a two bed maternity room [14, 15]. The health care system in Uganda is a referral system, with voluntary village health team members (VHTs) operating as the first contact point for patients. The next contact point is Health Centre II, which refers to Health Centre III or IV (District Hospital) if needed. The highest level of care is the National Referral and Teaching Hospital in the capital Kampala [3].

Based on their occupation and involvement in maternity issues, we recruited professional health workers such as midwives, students and nurses from the maternity ward at St. Francis Hospital and Kabizzi Health Centre. We enrolled mothers from the hospital or the villages, with the inclusion criteria that they had given birth within the past month. Traditional birth attendants (TBAs) were included based on their role as often being the first line contact with pregnant mothers in the villages, together with VHTs, who are the local representatives of the professional health care system in Uganda. We have displayed the data collection process in a flow chart below (Fig 1).

## Sampling

The villages were selected by approaching local village leaders, who lingered after conclusion of a village committee meeting in the urban centre of Nyenga town. We gave an introduction about the study and made arrangements to meet in their respective villages. In cooperation with the village leaders, the main researcher and two local research assistants enrolled a total of 57 participants in the study. Fifteen key informants were recruited for interviews, and 42 mothers attended focus group discussions. The key informants consisted of 4 mothers, 4 TBAs, 3 VHTs and 4 health workers. With exception of the professional health workers, all participants were small-scale subsistence farmers with some having additional small businesses. The education level ranged from none to certificate level for the health workers, as displayed in Table 1.

## Data collection

We used two semi-structured interview guides as instruments for the data collection, where one guide was directed towards mothers, and the other to key informants. The interview guides were locally pre-tested and amended before use in interviews with key informants and in focus group discussions (S1 and S2 Files).

Two local research assistants were trained and educated by the main researcher before initiation of the study. One male assistant was handling logistic organization, recruitment of key informants together with local village leaders and was moderator in focus group discussions.

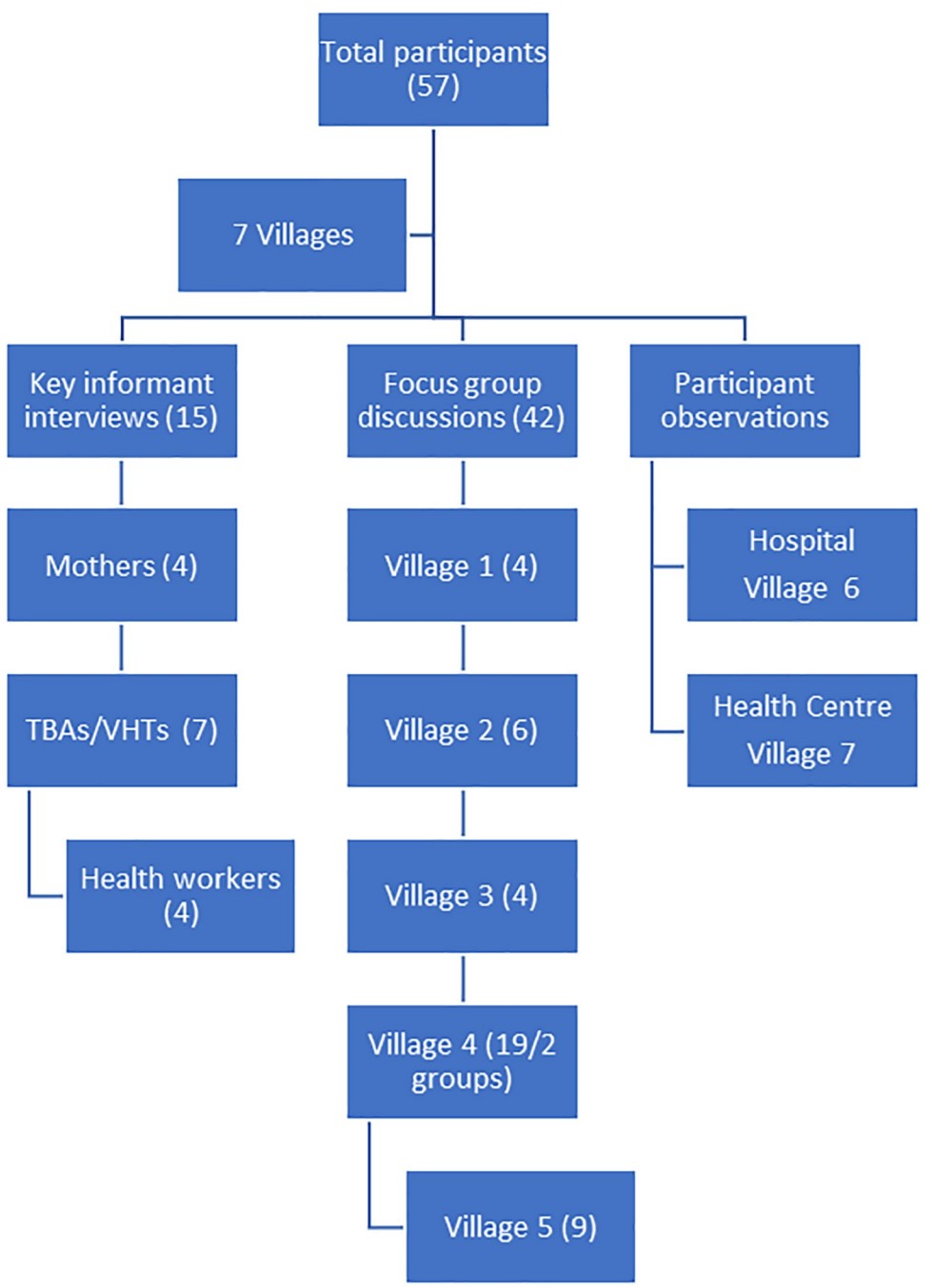

**Fig 1. Flow chart of data collection process.**

One female assistant was doing translations and transcriptions of recordings in the local language Luganda and had a role as moderator in focus group discussions. Both were fluent in English and Luganda with bachelor's degrees in social sciences, working within the same field.

The time period for data collection was from January 15th to February 25th, 2019.

**Key informant interviews.** We conducted semi-structured key informant interviews with 15 participants in quiet outdoor settings in the villages, or by the Hospital or Health Centre. The interviews were carried out by the main researcher in English ($n$ = 5), or by one of the

**Table 1. Characteristics of participants.**

| Participants | Key informant interviews | Focus group discussions | Age | Occupation | Education |
|---|---|---|---|---|---|
| Health workers (4) | 4 | | 19–35 | Nurses, midwives, students | Certificate level |
| Mothers (46) | 4 | 42 | 16–45 | Subsistence farmers, small scale business | Primary to secondary |
| TBA's (4) | 4 | | 50–80 | " | None to primary |
| VHT's (3) | 3 | | 50–80 | " | Primary |
| Total 57 | 15 | 42 | | | |

research assistants in Luganda, if the interviewee was not comfortable with the English language ($n = 6$). Upon request from the participants, one interview with key informants was a group interview including two TBAs and two VHTs, where each question was answered successively among them. The duration of the interviews ranged from 10 to 30 minutes.

**Focus group discussions.** Six focus group discussions were held in five villages, where the local leaders had provided outdoor sites in quiet areas under large trees. In one village, two discussions were held simultaneously due to the high number of mothers gathered. One of the local research assistants had the role as moderator in Luganda and gave intermittent brief summaries in English to the main researcher. She was present for observation and note taking during all focus group discussions, which lasted from 40–60 minutes.

**Participant observations.** The main researcher, who is a nurse by profession, spent six weeks observing daily routines and interactions between staff and mothers and their families in the Hospital and Health Centre. She engaged in unstructured dialogues with mothers and staff as a tool for understanding the dynamics of family and partner engagements, division of responsibilities within the health system and other support systems in maternal issues. Notes were written continuously.

## Data management

Transcriptions from audio-recorded interviews were typed into Microsoft Word documents within 2–4 days. If conducted in the local language, we transcribed the interviews first in Luganda, then into English consecutively. Final proofreading was performed by a third assistant, who listened to the audio while reading through the transcripts in both languages. Field notes from observations were typed upon completion of the study period.

All electronic data are kept behind two-level password protection in OneDrive. Notes and hard copies are stored in a locked cabinet and will be destroyed after 5 years from the time of data collection.

## Data analysis

The authors analysed the data based on the method called Systematic Text Condensation (STC), developed by Malterud [16]. Following the STC method, we read the transcribed text documents in-depth several times, in order to detect emerging themes from the raw data. After identifying the themes, the authors developed descriptive headlines and created coding trees using the NVIVO 12 pro software program. We pulled extractions from the transcribed documents and placed them under the most fitting code group. Several alterations of the coding structure were done during the process. Ultimately, the authors agreed on three themes created under the topic of "Maternal help and support", which were identified as: (a) Antenatal care and spousal support, (b) Health care and family support around delivery, (c) Postnatal care and challenges related to gender roles. The contents from the different code groups were

further analysed by creating condensed narratives supplemented with "golden quotes" to complete the descriptions.

The main researcher (MBR) was a female nurse from Norway who had visited Uganda several times previously. She was in charge of the primary data collection, coding and analysis of the data. IMSE assisted with co-reading the raw-data and discussion of main themes with the first author. RM contributed with developing the interview guides, acquisition of data and discussion of main themes with the first author. IN collected data materials, discussed findings and interpretations with the first author and helped with critical revision of the final paper.

In November of the same year, the participants in the study were invited for a dissemination meeting where they were presented with the findings and encouraged to modify, comment on, or question the results. The participants did not know the status of the other participants as informants. Feedback from the participants included a request to the local health workers and village leaders of a strategy to include and educate partners and men in maternity issues.

## Ethical considerations

The research was approved by the Regional Committee for Medical and Health Research Ethics, Norway (2018/602/REC West), Makerere University Higher Degrees Research and Ethics Committee, Uganda (HDREC/2018/6). Signed consent for internship/research was obtained from St.Francis hospital Nyenga and Kabizzi Health Centre. The study was registered with Uganda National Council for Science and Technology (HS302ES). Study participation was fully voluntary. Information about the study was given both in English and Luganda when necessary, and the participants signed or fingerprinted consent forms. During focus group discussions, each participant was given a number for recognition and was asked not to share private information gained during the sessions outside the group. Snacks and refreshments were offered to the participants before the interviews and focus group discussions, and they were reimbursed for transportation costs up to 15 000 UGS (equals 4 USD). The mothers attending focus group discussions received a piece of locally made baby clothing after the session and no extra money was given for participation.

## Results

One of the interesting aspects with conducting qualitative research is the emergence of views and shift in focus during interviews or focus group discussions. In this study, views about the importance of antenatal care and male involvement in maternal issues were given much attention both from mothers and TBAs, although they were not at the centre of the focus group discussion guides. Hence, the authors decided to give these topics augmented attention in this paper. Many women in the study were discontented with spousal support when they endeavoured to attend antenatal classes, which was often related to financial situations and transport challenges. The mothers had mainly positive experiences with professional health workers and TBAs around the time of delivery, where TBAs often played a key role. Postnatal care initiations from the professional health workers were found close to non-existent, except for the six-week check-up combined with the vaccination program. The mothers revealed disappointment with postnatal spousal support, such as unmet expectations of practical help, challenges related to gender roles, poverty, and inhibitions in own decision-making. Observations by the researcher could occasionally deviate from the view of the participants in the study and are conveyed explicitly. In the following section, the authors present the themes extracted from the analysis: (a) Antenatal care and spousal support, (b) Health care and family support around delivery, (c) Postnatal care and challenges related to gender roles.

## Antenatal care and spousal support

Among the interviewed mothers, knowledge of the importance of antenatal care was widespread, although many admitted non-attendance at antenatal classes for various reasons like road conditions, distance, and transport cost. Mothers explained how they were dependent on their partners for financial support, and how it could sometimes be difficult to convince them of the importance of antenatal care. Also, the partner could be away from home, which gave them no opportunity for traveling:

> *Sometimes a woman can request some money for transport from the husband and he tends to refuse to give out the money to the wife hence she ends up not going for antenatal care. That's why the husbands send their wives to go to these TBAs because for them they will not pay money.* (Mother 35–45 years, village 4)

The TBAs and VHTs brought up antenatal care as an important issue, and they stressed the significance of attending antenatal classes to gain information on hygiene and general baby care, referring to some mothers as "*totally green about some issues*". (TBA, village 7). They were especially concerned about very young mothers, sometimes only 14–15 years old, who did not have sufficient experience or knowledge about how to care for themselves or their newborns. The TBA's were advocating for a closer work relationship with professional midwives and health workers in the community, as a way to reach out to all new mothers. Some of the TBA's confirmed the statements of the mothers, that non-attendance in antenatal classes reflected the total financial situation in the family, where the lack of funds prohibited transportation to facilities.

## Health care and family support around delivery

For various reasons like rapid labour onset or going into labour after dark, the mothers did not always reach professional help in time for delivery. In one key informant interview, a mother explained in detail how she admired her husband for taking responsibility in asking a neighbour for assistance with delivering their baby, due to rapid labour combined with difficulties obtaining transport. Reliance on and support from family members in times of need and during vulnerable circumstances contributed to better coping and a sense of well-being.

On several occasions the researcher witnessed very young mothers who arrived with their mothers-in-law in time for delivering, both at the Hospital and the Health Centre. The researcher also witnessed many caring and supportive partners and husbands while present for observations at the Hospital and Health Centre. Men who accompanied their partners could be seen anxiously waiting for hours outside the maternity ward, awaiting the birth of their baby son or daughter, or they travelled long distances to reach the delivery.

Contrary to the mentioned statements and observations, many women who participated in focus group discussions showed resentment and anger towards negligent husbands and fathers, regarding unmet expectations of help and support in time of labour, as well as on economic issues:

> *A very big number of women from this village give birth from their homes just because they lack money and even the husbands do not fulfil their responsibilities, which sometimes leads to both infant and maternal mortality.* (Mother 35–45 years, village 4)

Being at a health centre with marginal resources and equipment when complications occurred, was described as a huge challenge for the health workers, and they often found

themselves alone on duty. When there was need for referrals, it was the family's responsibility to arrange for transport, but that was often described as difficult and unreliable. One midwife interviewed had a traumatic experience freshly in mind of how she struggled to find transport for referral of a mother who faced complications during labour, implicitly also describing her own support to the mother and her family:

*I was alone, and the baby was at the outlet, I was seeing the head actually. I tried several boda-boda names (scooter driver's names), but their phones were off. . .before reaching (village 1) we got in an accident around the house near N. Then we reached the road, it was raining, totally raining, and the petrol got finished. And she was fitting (having seizures) with heavy rain. We stayed in (village 1) up to morning. They had to give magnesium, but it took long for her to give birth. She delivered when she was still fitting. They did episiotomy when she was still fitting.* (Midwife 19–35 years)

### Postnatal care and challenges related to gender roles

Due to the various components of postnatal care, the theme will be presented with sub-topics as follows; postnatal follow-ups, breastfeeding and nutrition, family planning, and gender inequality. The sub-topics will be combined in the discussion.

**Postnatal follow-ups.** Pre-scheduled or planned postnatal follow-ups before vaccination of newborns at six weeks was not common or recognized among the professional health workers. When inquiring from the health workers about postnatal follow-ups or home visits, the general answer was that they told the mothers to return to the facility if they should face any challenges, but they also confirmed that this seldom occurred. Newborn vaccinations against polio and BCG were normally given at the Hospital or Health Centre before discharge.

For women giving birth at home, the TBAs or VHTs could offer advice to go to the hospital for vaccinations and measurements, but it was the mothers' responsibility to follow up. Some of the TBAs interviewed took extra measures to follow up on the mothers they had helped with deliveries:

*Yes, I do visit them after some time from the day I discharged them to find out how they are doing, if they are in good condition both the mother and the baby.* (TBA 50–80 years, village 1)

Others explained how they recommended them to seek postnatal care and go for vaccinations at various health facilities.

**Breastfeeding and nutrition.** The midwives were often alone on duty, and due to hectic work environments and sometimes attending to several mothers in labour simultaneously, the information and support for breastfeeding were thus neglected. Sometimes the mothers were fortunate to have gained experiences through family ties.

Making sure that the mothers initiated breastfeeding early were important to the TBAs, and when mothers faced challenges of sore nipples, they would sometimes go to extreme measures to make the mothers breastfeed:

*There are some mothers when you tell her to breastfeed the baby, she can hesitate [saying] that she feels nipple pain, me I even slap some of them for hesitating.* (TBA 50–80 years, village 1)

**Family planning.** Women's autonomy and own decision-making were challenged in reproductive health issues. Recurrent topics raised by women attending focus group

discussions were sexual activity after birth and family planning, where the discord between the various needs of men and women were repeatedly proclaimed, and, as the following quote shows, it could prove a risky affair for the women:

> *If a woman goes for family planning on her own and the man gets to know it, it will just become a fight or even he can kill the wife.* (Mother 35–45 years, village 4)

From observations at the Hospital, the researcher witnessed a young mother who had requested tubal legation after having had four Caesarean sections. The procedure had been recommended and approved by the doctors for the health of the mother and her husband had signed the consent papers. On the day of the procedure the husband withdrew his approval, and the mother was not allowed to go through with the surgery.

Unstructured dialogue with mothers during observations revealed that many were aware of other contraceptives like intra-uterus devices (IUDs), but intimidating stories of painful insertions and side-effects of infertility prohibited usage.

**Gender inequality.** Whether one was a first-time mother or had other children, women described the early postnatal period as challenging due to sleep deprivation, breastfeeding problems, daily chores, and uncertainties about motherhood. Unmet expectations from their partners resulted in exposure of underlying feelings of being neglected and betrayed, which became transparent in this quotation from a focus group discussion:

> *Me*, *sometimes after giving birth I feel like eating posho (*local dish from maize*), but my husband tends to run away from his responsibilities and goes and marries other women*, *me I even fetch water for myself. (Mother, 25–35 years, village 4)*

During the observation period, the researcher witnessed supportive husbands or other family members who cared for the mothers after they had given birth. Single mothers would often be accompanied by a sibling of the mother, who would assist her in caring for the newborn, or fetching food and water for washing. However, the negative experiences did influence the interview data.

## Discussion

The initial platform to ensure safe quality care for pregnant women and their unborn babies is antenatal care visits. The presented results indicate that antenatal care was not always accessible for all mothers in Buikwe district, due to distance and transport costs as the biggest obstacles. The women relied heavily on their partners and their support to access the services. WHO/UNICEF Uganda recommends eight antenatal visits during pregnancy, whereas the UCG aim for at least 4 visits [2, 4]. Studies from sub-Saharan Africa have shown clear associations with attending antenatal care and reduction in neonatal mortality [17], even though timing and frequency for antenatal visits vary among regions and socioeconomic status [18]. Studies from Uganda concerning availability and quality of antenatal care provision in rural settings show lack of qualified staff and inadequate check-ups and provision of necessary information as key areas for improvements [19]. One study found that sampling of urine and receiving drugs for intestinal parasites were components most often neglected during antenatal visits, and that the overall quality of antenatal care has been found to be higher in private sector facilities than in public ones [20]. Intervention programs on improving quality of care show promising results, but reveal the need for systemic improvements in infrastructure and education of health care providers [19]. Statements from TBAs and VHTs in this study, reveal concerns about some young women's low knowledge about issues like hygiene and how to care

for newborns. This indicates a need for closer cooperation between TBAs/VHTs and professional health workers in rural Uganda.

Both mothers and TBAs referred to poverty as the key reason why women were not able to attend antenatal care. The same factor can possibly also explain low male involvement in maternity care, since the transport cost would double if two people were to travel instead of one, in addition to income and working time lost. The importance of supportive environments and recognition of partner involvement interventions in maternal health, has gained increased attention since the introduction of WHO's *Maternal and Child Health Care Program* (MCH) in the mid 1990's [21]. The multi-lateral global movement *Every Woman Every Child* includes strategies of male involvement in maternal health programmes as one of its action areas towards achieving the SDGs before 2030 [22]. Focus on male involvement in maternal care is further strengthened in the national strategy for *Sexual and Reproductive Health and Rights* (SRHR), which was launched by the Ugandan Government in 2014 [23]. Male involvement in maternal care is also implemented in the UCG by recommending bringing the partner or a family member to at least one antenatal visit [2]. Previous studies from Uganda have confirmed that the spouse often remains at home looking after the household and other children, which allows the woman to go for antenatal care [24]. A documented cost-effective way to increase male attendance is to give a personal invitation letter to the women's spouses, as an alternative to a general information pamphlet. This intervention has proven to raise the attendance by up to 10% [25], although it does not solve the underlying economic problem.

Traditional and cultural customs of men being unwelcome in the delivery rooms could contribute to signals of men not being wanted or needed in situations around childbirth [26]. Looking at the situation from another angle, the men might not always have a choice of attending to their wives based on work conditions and availability. Given time off work to tend to one's wife and child around the time of childbirth is constituted in the Employment Act of Uganda from 2006, which gives fathers four days paid leave from work to spend with the family [27]. However, many men do not have regular work conditions to effectuate that. Obstacles to male involvement in maternity care are grounded in both culture and religion, as shown in studies within Nigerian and Ugandan settings, where women tend to share similar views as men when it comes to what is considered appropriate and expected, although it varies within the socioeconomic strata [28, 29]. Many mothers in the study did not have safe environments around the time of childbirth. Challenges related to logistics and economy were explained as reasons for suboptimal birthing experiences, both from the user and provider perspectives. Weather and road conditions reflect the vulnerability of the health system in rural Uganda and add to the challenges of transport costs and low staffed health facilities. Hence, many women must succumb to the environment and give birth in the villages with help from TBAs.

Postnatal check-ups are recommended in the UCG at 6 hours after birth, after 2–7 days and after six weeks, and include counselling on emergency issues, hygiene and general baby care [2]. Neither the health workers nor the mothers were familiar with the UCG recommendation of 2–7-day check-ups [2], which would give the mothers a good platform for expression of concerns around breastfeeding issues or other postnatal conditions. Some of the health workers interviewed recommended mothers to come back if they faced any trouble post-partum, but that was not routine. The recommended level of professional support in the postnatal period has changed over the years, but recent WHO guidelines give higher attention to this vulnerable time in a mother's and her baby's life. The new guidelines include postnatal care on the first day, third day, between 7–14 days and six weeks [8], but these recent recommendations are not yet included in the UCG. Postnatal contact between mothers and TBAs was more common, and a study from Nigeria show that both monetary and non-monetary payments to TBAs gave incentives for referral of mothers to health facilities [30].

A feeling of security and supportive surroundings are requirements for a good bonding- and early breastfeeding experience, and the lack of such may lead to post-partum depression and trouble with, or discontinuation of, exclusive breastfeeding [31]. Emotional wellbeing and family support is equally important [8]. Midwives reported challenges with time constraints as factors for not providing sufficient information and support in breastfeeding issues for the mothers. This study points towards shortcomings both in frequency and quality of postnatal encounters between health workers and mothers.

Some mothers showed resentment towards their spouses for not assisting them during and after birth and uttered disappointment with spousal neglect in providing them with nutritious food and support with household chores they normally were doing. Stories told with anger and bitterness reflected the hopelessness and despair many women found themselves to be in. Some of the mothers mentioned sharing concerns and advice with neighbours and friends, but when it came to provision of physical help it often seemed to be each woman for herself. The resentment was coupled with aversion against early sexual demands. In the Ugandan culture, many women are subjected to the wants and decisions of men, as described clearly in an article in *The Observer* by Kiiza and Akumu [32]. Although the Uganda Children's Act deems the parents equal [33], culturally the children are seen as the property of the father, and the mother of the child cannot deny the father sexual favours [32]. In addition to the feeling of lack of support and care, women also face the problem of being economically dependent on their partners, enhanced by legislative regulations that favour males [34].

The mothers interviewed in this study complained about unreliable financial support from their husbands and made them partly responsible for the poverty. This study did not capture fathers' views on the situation, however, other studies from Uganda describe gender disparities as the causal factor of poor access to sexual and reproductive services [24]. Additionally, men's lack of financial support in maternal health could often be justified by the overall lack of financial resources at home [26]. The women's discontent with support from their partners could be an indication of alterations of traditional gender roles in view of recent years' access to internet services and influences from social media [35].

Findings from the presented study, related to women's autonomy in reproductive health issues mirror similar findings concerning maternal health and support systems in Uganda [26]. Maternal and newborn health research could continue with gender sensitive perspectives, taking into account disparities that may arise from the presence or lack of family support in vulnerable pregnant and delivering women, and for the mother and baby in the postnatal period.

## Study strengths and limitations

The authors sought to obtain reliability by using local translators and trained interviewers of both sexes. Possible bias includes that the main researcher and primary analyst was a foreign person and holds an etic, meaning subjective, view of the situation witnessed in Uganda during the research period. Alternatively, sometimes being from a different country or culture can have a positive effect on the interviewees as it is seen as less threatening, as someone from the same culture may be more prone to criticize local practices. Also, the supportive statements and validation of data from local women give extra strength to the observations from the study. A majority of the negative statements about partners' support on maternal issues were from the same focus group discussion, where many women were gathered. Being many may have encouraged others who otherwise would have kept silent. The study did not include interviews with fathers or partners, which could be seen as limitations to the study. Including the voice of partners could have given a more nuanced picture of the situation reflected in the

study, and a better understanding of their situation. Proofreading of transcriptions added credibility to the study. Issues with electric power and technical difficulties resulted in written notes only, from one focus group discussion and one in-depth interview. Dependability of the study was sought obtained through triangulation of qualitative methods, although no participant observations were done outside of the health facilities. Previous research results from rural settings in Uganda makes the study transferable to settings with similar context and clienteles.

## Conclusions

Maternal support and care were highly sought and valued by the mothers in the study, but not always accessible due to logistic and financial problems. Resources were scarce both in the homes and within the health system. Further focus on gender equity on all policy making levels, involving women's right to own decision making in maternity issues and higher recognition of male involvement in maternity care, are suggestions to policy makers for improved maternal health and newborn survivals in rural Uganda. This study highlights a continued need for higher awareness and incentives for implementation of recent WHO recommendations on postnatal care in the combat against infant mortality.

## Supporting information

**S1 File. Interview guide for key informants.**
(DOCX)

**S2 File. Interview guide for focus group discussions and mothers.**
(DOCX)

## Acknowledgments

The authors wish to thank the participants in the study for their interest and cooperation. Authors also extend gratitude to Nyenga Foundation Norway/Uganda, and School of Public Health Makerere University, Uganda and Centre for International Health/ University of Bergen, Norway Collaboration 2018–2021.

## Author Contributions

**Conceptualization:** Marte Bodil Roed, Ingunn Marie Stadskleiv Engebretsen.

**Data curation:** Marte Bodil Roed, Robert Mangeni, Irene Namata.

**Formal analysis:** Marte Bodil Roed.

**Investigation:** Marte Bodil Roed, Robert Mangeni, Irene Namata.

**Methodology:** Marte Bodil Roed.

**Project administration:** Marte Bodil Roed.

**Supervision:** Marte Bodil Roed, Ingunn Marie Stadskleiv Engebretsen.

**Validation:** Marte Bodil Roed, Ingunn Marie Stadskleiv Engebretsen, Robert Mangeni, Irene Namata.

**Writing – original draft:** Marte Bodil Roed.

**Writing – review & editing:** Marte Bodil Roed, Ingunn Marie Stadskleiv Engebretsen, Robert Mangeni, Irene Namata.

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
