## [Decision Letter · Decision Letter 0]

4 Aug 2021

PONE-D-21-14606

Women’s access to and use of maternal health care and home support systems to reduce neonatal deaths in Buikwe district, Uganda: a qualitative study

PLOS ONE

Dear Dr. Rød,

Thank you for submitting your manuscript to PLOS ONE. After careful consideration, we feel that it has merit but does not fully meet PLOS ONE’s publication criteria as it currently stands. Therefore, we invite you to submit a revised version of the manuscript that addresses the points raised during the review process.

We look forward to receiving your revised manuscript.

Kind regards,

Calistus Wilunda, DrPH

Academic Editor

PLOS ONE

Journal Requirements:

Reviewers' comments:

Reviewer's Responses to Questions

**Comments to the Author**

1. Is the manuscript technically sound, and do the data support the conclusions?

Reviewer #1: Yes

Reviewer #2: Yes

Reviewer #3: Partly

2. Has the statistical analysis been performed appropriately and rigorously? 

Reviewer #1: I Don't Know

Reviewer #2: N/A

Reviewer #3: N/A

3. Have the authors made all data underlying the findings in their manuscript fully available?

Reviewer #1: Yes

Reviewer #2: No

Reviewer #3: No

4. Is the manuscript presented in an intelligible fashion and written in standard English?

Reviewer #1: Yes

Reviewer #2: Yes

Reviewer #3: No

5. Review Comments to the Author

Reviewer #1: Thank you for inviting me to review this manuscript on women’s access to and use of maternal health care and home support systems in Uganda. I have enjoyed reading the paper as it speaks to a topic close to my own scholarly interests and has numerous quotes that point to some original and important qualitative data.

Some of my major comments and constructive criticisms relate to 1) the structure, organization, and language of the paper, 2) coverage of appropriate literature, and 3) adequacy of analysis, interpretation, and definition of key concepts, terms. For example, in the title of the manuscript it says, “to reduce neonatal deaths,” but the qualitative data and discussion is less about neonatal deaths and more about structural barriers to accessing biomedical, facility-based health care such as expenses, transport, social and economic support from spouse. I would encourage the authors to think a little more about the main ideas that they would like to put forth in the paper and restructure/organize the data to support 1-2 main arguments instead of summarizing the findings. There are a lot of great data and analysis in the discussion but some of the this gets lost in the way the paper is currently organized. Overall, there are too many quotes used throughout the paper. PLOS One has a wide audience and readership and you need to be able to walk each reader through your paper by providing enough background, context, and utilizing only the quotes that speak directly to your main argument. We do not know how or why a quote is important if you do not articulate your main argument and relate the quote back to this idea. Perhaps restructuring the data and discussion sections so that the discussion comes after each of the different thematic data sections. You could also use the perspectives of mothers, TBS, and health workers as the main data sections to try to show the variation in the themes that were important to each group. Or if you keep the current structure there needs to be more specific analysis in each data section as right now seems to be a grouping of quotes that do not all seem to fit under the thematic heading. I also put “I don’t know” in regards to the question about the statistical analysis because the way the data is currently structured I cannot say if the Nvivo analysis was performed appropriately and rigorously.

I would also encourage the authors to go through and try to make writing more active throughout the manuscript by getting rid of passive voice and reducing the number of words that end with “-ing.” For example, in the Introduction, line 45, it says, “Uganda is a country struggling with reducing” and this could become “Uganda is a country which has struggled to reduce…” Same thing in the abstract, line 28 could become “This study aimed to describe…” instead of “aimed at describing.”

I think there also could be additional inclusion of other social science and qualitative literature. There has been a lot of anthropological literature published on this topic in Uganda and other countries in sub-Saharan Africa that could be useful for adding context. Helle Max Martin (Nursing Contradictions) has written about maternal health in Uganda and also look at recent publications in BMC pregnancy and birth, Social Science & Medicine, Health Policy & Planning.

Here are some additional comments and questions:

Lines 49-50, Can you be more specific about what you mean as “good quality maternal health care and professional support systems”? Perhaps offer some citations here. What constitutes good quality—how is this being defined, etc.?

Lines 76-78, Can you describe this more? I am not sure how giving a written letter to a spouse would increase male attendance by 10%. Who writes the letters? Are their fines or sanctions for not attending if one receives a written letter?

Line 100, Can you offer a definition for how you are conceptualizing/defining support? There are lots of types of support—such as economic, social, emotional, etc. I think offering a brief definition about how you define/use maternal support as a concept would be helpful here.

Line 130, Passive voice—try to rephrase. Also making it active would allow reader to see who was doing this recruitment which is a question I have. Who recruited participants for the study? The authors, village leaders, etc?

Line 132, You are using acronyms, but have you already defined them? Need to spell them out first and add additional context about who these people are and the roles the perform.

Line 142, Okay here I see there were local researchers helping with recruitment and data collection. This may be helpful to bring up with the previous section or mention this earlier.

Lines 157-158, How did you determine whether to interview someone in English or Luganda? By their wishes, proficiencies? Who made that call? Perhaps you could explain more briefly.

Lines 179-198, this section, “Data analysis” seems to be entirely in passive voice. I think switching into active voice enables clearer, stronger writing. Is there a way you can rewrite, even if it brings you into this more? You can say, “Authors analyzed data using the STC method developed by Malterud.

Lines 226-227, What about other non-biomedical forms of postnatal care? I work in Tanzania, where women access many different forms of care, from family members, TBAs, etc. Perhaps if you are going to state that postnatal care was non-existent specify what type of postnatal care, such as facility-based/biomedical, you are referring to here.

Lines 308-310, This quote from data does not seem to be analyzed or unpacked, overall, I would cut down on quotes and only include the ones that directly relate to your argument about maternal support, which you also should clearly state more thoroughly. You are offering a summary of findings but I as a reader am starting to get a little lost about what you are doing with this summary—where you are going and the significance, meaning of it all. I think cutting back on quotes would help and you need to offer analysis, context, for each quote you use, relating it back to your main ideas or arguments.

Reviewer #2: Review assignment for PONE-D-21-14606

TITLE: Women’s access to and use of maternal health care and home support systems to

reduce neonatal deaths in Buikwe district, Uganda: a qualitative study

Summary

This is a good study that provides interesting qualitative findings of the access and use of women of maternal support systems to reduce neonatal mortality. Key themes emerged from this study highlighting financial and logistical problems that led to inadequate maternal support and care. However, the excerpts from the interviews highlight how important these voices are in amplifying and highlighting key issues emerging from this research.

Key issues to consider

'

Introduction

There is a lot of focus on antenatal care in the background section but the relevance of all the information provided is not coming out clear in the paper

Would be good to have a description of what the home support systems entail in the background so that one can be able to know what the study is all about such that each component is unpacked in a way that flows in the background of the paper

It would have been good to see how these support systems have worked or been implemented in different contexts besides the African context as has been presented based on evidence from the literature

The Ugandan clinical guidelines are mentioned in the background line 75. Some brief descriptions of what they are and how relevant they are for this study.

This section could also incorporate aspects that come up in the discussion including gender, culture and socioeconomic disparities that would

Methods

These have been well described although some sections could be summarized to give information that would be important to report in a paper rather than the full details of the specific steps taken. This can be done with guidance from the editors.

There is a need to describe the context a bit more so that one can be able to understand and relate some of the specific findings of these populations that could be linked to the culture and available systems within this context to support maternal care and postnatal care.

Results

The themes have been summarized well, although one who has more experience in reviewing qualitative research could provide more insight into the statistical analyses and results that have been described.

The key themes are supported by the reported findings from the participants.

“the importance of antenatal care and expectations to fathers were given much attention both from mothers and traditional birth attendants, although not being a key focus of the topic guide” – There is a lot of discussion in the background about ANC so why did the authors choose that it is not a key focus of the topic guide. I also wonder about spousal support which is very key when it comes to maternal care and pregnancy.

“Many women in the study were discontent with spousal and family contributions related to help and support in maternal issues, often related to pecuniary difficulties” Not clear what this sentence means

The issue of gender comes out in the results but is not well described in the background section of the paper

Line 306 and 307 could be moved to the health worker section as it is not linked to the gender roles section

The section on Postnatal care and challenges related to gender roles. Could be reorganized to bring out key themes to improve flow for example i. gender inequality ii. breastfeeding and nutrition, iii. Family planning iv. Postnatal care etc.

The discussion on spousal support this section could also be moved to the relevant section

Discussion

This statement in line 361 needs to be re-written as it comes out as a strong statement likely just due to the wording “not a well-established practice for all mothers in Buikwe district.”

This comes out in the discussions but is not adequately discussed in the results “young women’s low knowledge about issues like 381 hygiene and how to care for newborns”

“Challenges related to logistics and economy were explained as reasons for suboptimal experiences both from the user and provider perspective.”- This needs to be further explained as it is not clear and also a strong statement

The UGC comes up again in the discussion but is not well discussed in the background/ methods so that one can be able to relate to it in the discussion

Minor comments

Some terminologies that come out as a bit strong making some statements difficult to interpret

“In acquiring to understand” change wording here

Line 132 “TBAs, 3 VHTs” write in full

Discrepancies in the use of abbreviations throughout the document for example as above and in the results and discussion sections

Reviewer #3: Overall comments:

This is an important piece of work that is addressing a key global health priority. However, the themes reported on maternal experiences of health care are very similar to what has been found in other contexts in sub-Saharan Africa, so it is ever so important that the authors draw out what their key messages are and what this adds to the literature. They also need to link it better to neonatal outcomes as they have stated in the title. Unfortunately, most sections are not written at the level required for a publishable manuscript. Some of the sections are very long and lack focus (e.g. Introduction and Results). The Methods section needs to be reorganised and, in some sections, explained in more depth. The whole manuscript would benefit from a review of the grammar.

Title: Women’s access to and use of maternal health care and home support systems to

reduce neonatal deaths in Buikwe district, Uganda: a qualitative study

I appreciate the complexity of what was being explore in this study but suggest the title captures this work more succinctly. For example:

Women’s experiences of maternal and newborn health care services in Buikwe district, Uganda: a qualitative study

Abstract

Background: This study aimed at describing maternal support systems – revise to This study aimed to describe….

Methods and findings:

Data Management is missing.

Line 73…prohibited women in receiving optimal – replace in with from.

Line 38 Unclear how “ Postnatal follow-ups were found unsatisfactory.” This needs to be explained a bit more in the Abstract as it is linked to a recommendation in the Conclusion.

Please review the grammar in the Abstract.

Introduction

This section needs to be a lot more succinct and linked to the aims of the study. As written, it is too long. Some of this content can be moved to the Discussion.

Some specific points:

Line 47 … 2030 if today’s trend continues- revise to: if the current trend continues.

Line 51-53…. For the mothers it is also important to have someone to trust and rely on during pregnancy, time of childbirth and the following post-partum period, and by such providing an optimal start in life for the newborns (3). So that…please expand on this.

Methods

Line 108 …individual interviews – I am not clear what individual interviews are because in qualitative research, interviews are conducted between a researcher and a participant and not a group. Please clarify.

Line 108…key-informants- do you mean key informant interviews?

Line 113- A summary of methods is given below: please delete this as it doesn’t provide any additional

Line 127- Participants were selected purposefully – based on what characteristics? Please expand. (mentioned in line 149-150)

Line 129- Sample – This subtitle should be “Sampling”. It would be useful to have a Table linked to this section to summarise the sampling strategy.

Line 176 Instruments This should be incorporated at the beginning of the sub-section: Data Collection.

Data Management is missing.

Results

Introductory paragraph needs to be revised as it doesn’t lead the reader well into the content of the results. This section is also long so the authors need to decide what the key messages are and re-write this in a logical manner that brings out the key messages.

Discussion

This section of the manuscript is more structured with clearer messages of the relevance of this work in the context of existing literature on the topic and context. However, the focus on maternal experiences doesn’t link adequately to neonatal outcomes as stated in the title so I wonder whether the paper should focus on maternal experiences of accessing health care.

Conclusion

No comments

6. PLOS authors have the option to publish the peer review history of their article (what does this mean?). If published, this will include your full peer review and any attached files.

Reviewer #1: No

Reviewer #2: No

Reviewer #3: No

---

## [Author Response · Author response to Decision Letter 0]

13 Sep 2021

PONE-D-21-14606

Women’s access to and use of maternal health care and home support systems to reduce neonatal deaths in Buikwe district, Uganda: a qualitative study

PLOS ONE

Dear Editors and Reviewers at PLOS ONE,

The authors thank you for considering the manuscript for publication in your journal, and for your thorough and constructive feedback. We have made our best effort to adhere to your recommendations and hope the manuscript is now further qualified and organized in line with the requirements of PLOS ONE. A point-by-point response to the editor and reviewers’ comments is given below. 

Sincerely, on behalf of the authors, 

Marte Bodil Roed. 

Journal Requirements:

A revision of file naming has been conducted, and the title page edited according to the template of PLOS ONE. 

The above bullet points have been carefully revised and handled.

Line 221: Ethics statement has been included in the manuscript

Reviewers' comments:

Reviewer's Responses to Questions

Reviewer #1: Thank you for inviting me to review this manuscript on women’s access to and use of maternal health care and home support systems in Uganda. I have enjoyed reading the paper as it speaks to a topic close to my own scholarly interests and has numerous quotes that point to some original and important qualitative data.

Some of my major comments and constructive criticisms relate to 1) the structure, organization, and language of the paper, 2) coverage of appropriate literature, and 3) adequacy of analysis, interpretation, and definition of key concepts, terms. For example, in the title of the manuscript it says, “to reduce neonatal deaths,” but the qualitative data and discussion is less about neonatal deaths and more about structural barriers to accessing biomedical, facility-based health care such as expenses, transport, social and economic support from spouse. I would encourage the authors to think a little more about the main ideas that they would like to put forth in the paper and restructure/organize the data to support 1-2 main arguments instead of summarizing the findings. 

The authors agree and appreciate the feedback. The title and aim of the paper have been altered to focus more specifically on women’s experience on maternal and newborn health care services. Also, the data has been more compressed and hopefully more focused. 

There are a lot of great data and analysis in the discussion but some of the this gets lost in the way the paper is currently organized. Overall, there are too many quotes used throughout the paper. PLOS One has a wide audience and readership and you need to be able to walk each reader through your paper by providing enough background, context, and utilizing only the quotes that speak directly to your main argument. We do not know how or why a quote is important if you do not articulate your main argument and relate the quote back to this idea. Perhaps restructuring the data and discussion sections so that the discussion comes after each of the different thematic data sections. You could also use the perspectives of mothers, TBS, and health workers as the main data sections to try to show the variation in the themes that were important to each group. Or if you keep the current structure there needs to be more specific analysis in each data section as right now seems to be a grouping of quotes that do not all seem to fit under the thematic heading. I also put “I don’t know” in regards to the question about the statistical analysis because the way the data is currently structured I cannot say if the Nvivo analysis was performed appropriately and rigorously.

Thank you for this useful advice. We have restructured the data- and discussion section so that the discussion follows directly after each thematic data section. The quotes have been reduced. We hope that the manuscript now has a better flow and structure, and that the method of analysis is easier recognized in the paper. 

I would also encourage the authors to go through and try to make writing more active throughout the manuscript by getting rid of passive voice and reducing the number of words that end with “-ing.” For example, in the Introduction, line 45, it says, “Uganda is a country struggling with reducing” and this could become “Uganda is a country which has struggled to reduce…” Same thing in the abstract, line 28 could become “This study aimed to describe…” instead of “aimed at describing.”

Thank you for this suggestion. We have reduced on the use of passive voice and words ending with -ing. 

I think there also could be additional inclusion of other social science and qualitative literature. There has been a lot of anthropological literature published on this topic in Uganda and other countries in sub-Saharan Africa that could be useful for adding context. Helle Max Martin (Nursing Contradictions) has written about maternal health in Uganda and also look at recent publications in BMC pregnancy and birth, Social Science & Medicine, Health Policy & Planning.

Line 53: A deeper understanding of the underlying factors of neonatal mortality in Uganda have been added in the introduction section. 

Here are some additional comments and questions:

Lines 49-50, Can you be more specific about what you mean as “good quality maternal health care and professional support systems”? Perhaps offer some citations here. What constitutes good quality—how is this being defined, etc.?

Line 56: WHO’s definition has been implemented and cited here. 

Lines 76-78, Can you describe this more? I am not sure how giving a written letter to a spouse would increase male attendance by 10%. Who writes the letters? Are their fines or sanctions for not attending if one receives a written letter?

Line 299: This paragraph has been moved to discussion under “Antenatal care and spousal support”, and explained in more details. 

Line 100, Can you offer a definition for how you are conceptualizing/defining support? There are lots of types of support—such as economic, social, emotional, etc. I think offering a brief definition about how you define/use maternal support as a concept would be helpful here.

Line 65: We have further clarified and defined our use of the maternal support concept in introduction

Line 130, Passive voice—try to rephrase. Also making it active would allow reader to see who was doing this recruitment which is a question I have. Who recruited participants for the study? The authors, village leaders, etc?

Line 140: It was the main researcher and two local assistants. The information has been specified and inserted here, and also rephrased to active voice. 

Line 132, You are using acronyms, but have you already defined them? Need to spell them out first and add additional context about who these people are and the roles the perform.

Acronyms and context of TBAs and VHTs have been spelled out and elaborated. 

Line 142, Okay here I see there were local researchers helping with recruitment and data collection. This may be helpful to bring up with the previous section or mention this earlier.

Has now been mentioned earlier. 

Lines 157-158, How did you determine whether to interview someone in English or Luganda? By their wishes, proficiencies? Who made that call? Perhaps you could explain more briefly.

Line 166: They were interviewed in Luganda if they were not comfortable with English language (have been specified). 

Lines 179-198, this section, “Data analysis” seems to be entirely in passive voice. I think switching into active voice enables clearer, stronger writing. Is there a way you can rewrite, even if it brings you into this more? You can say, “Authors analyzed data using the STC method developed by Malterud.

Thank you for pointing this out. The section is now more active. 

Lines 226-227, What about other non-biomedical forms of postnatal care? I work in Tanzania, where women access many different forms of care, from family members, TBAs, etc. Perhaps if you are going to state that postnatal care was non-existent specify what type of postnatal care, such as facility-based/biomedical, you are referring to here.

Line 246: Yes, this is true. It is now specified. 

Lines 308-310, This quote from data does not seem to be analyzed or unpacked, overall, I would cut down on quotes and only include the ones that directly relate to your argument about maternal support, which you also should clearly state more thoroughly. 

Thank you. This quote has been removed from the manuscript

You are offering a summary of findings but I as a reader am starting to get a little lost about what you are doing with this summary—where you are going and the significance, meaning of it all. I think cutting back on quotes would help and you need to offer analysis, context, for each quote you use, relating it back to your main ideas or arguments.

Thank you for this observation. The quotes have been reduced and the summary of findings has been re-written as is hopefully more focused and organized now. 

Reviewer #2: Review assignment for PONE-D-21-14606

TITLE: Women’s access to and use of maternal health care and home support systems to

reduce neonatal deaths in Buikwe district, Uganda: a qualitative study

Summary

This is a good study that provides interesting qualitative findings of the access and use of women of maternal support systems to reduce neonatal mortality. Key themes emerged from this study highlighting financial and logistical problems that led to inadequate maternal support and care. However, the excerpts from the interviews highlight how important these voices are in amplifying and highlighting key issues emerging from this research.

Thank you for appreciating the study and for helpful corrections and revisions of the paper.

Key issues to consider

Introduction

There is a lot of focus on antenatal care in the background section but the relevance of all the information provided is not coming out clear in the paper

Lines 257-302: The importance and relevance of antenatal care has been elaborated under the topic of “Antenatal care and spousal support”. 

Would be good to have a description of what the home support systems entail in the background so that one can be able to know what the study is all about such that each component is unpacked in a way that flows in the background of the paper

Lines 65-86: We have defined a clearer view on what is understood by maternal support systems, both professional, and community- and home based.

It would have been good to see how these support systems have worked or been implemented in different contexts besides the African context as has been presented based on evidence from the literature.

Line 84: A study on maternal support benefits for both mother and child in a European setting has been added to the context. 

The Ugandan clinical guidelines are mentioned in the background line 75. Some brief descriptions of what they are and how relevant they are for this study.

Line 51-53: We have added more details about the guidelines. 

This section could also incorporate aspects that come up in the discussion including gender, culture and socioeconomic disparities that would

Lines 80-84: More background and references on gender disparities and socioeconomic aspects have been added. 

Methods

These have been well described although some sections could be summarized to give information that would be important to report in a paper rather than the full details of the specific steps taken. This can be done with guidance from the editors.

This section has been shortened and restructured. If the section is still too specific, your guidance will be appreciated. 

There is a need to describe the context a bit more so that one can be able to understand and relate some of the specific findings of these populations that could be linked to the culture and available systems within this context to support maternal care and postnatal care.

Lines 113- 125: A paragraph about demographics of Uganda/Buikwe is now added. We also elaborated on the Ugandan health care system. 

Results

The themes have been summarized well, although one who has more experience in reviewing qualitative research could provide more insight into the statistical analyses and results that have been described.

The key themes are supported by the reported findings from the participants.

“the importance of antenatal care and expectations to fathers were given much attention both from mothers and traditional birth attendants, although not being a key focus of the topic guide” – There is a lot of discussion in the background about ANC so why did the authors choose that it is not a key focus of the topic guide. 

The topic guide was made beforehand as a tool included in the study proposal which focused more on delivery, breastfeeding and postnatal maternal support systems. Discussions around antenatal care arose in connection with questions related to place of birth, where women talked about giving birth where they had gone for antenatal classes due to reduced payments for services. The unexpected turns a qualitative study can take during data collection have been tried to explain in the result section. (Lines 237-242).

I also wonder about spousal support which is very key when it comes to maternal care and pregnancy.

“Many women in the study were discontent with spousal and family contributions related to help and support in maternal issues, often related to pecuniary difficulties” Not clear what this sentence means

Line 242: The sentence has been rewritten and is hopefully clearer. 

The issue of gender comes out in the results but is not well described in the background section of the paper

Lines 70-86: We have inserted a paragraph relating to previous research about gender aspects in the background. 

Line 306 and 307 could be moved to the health worker section as it is not linked to the gender roles section

After some consideration, this paragraph is now omitted from the paper.

The section on Postnatal care and challenges related to gender roles. Could be reorganized to bring out key themes to improve flow for example i. gender inequality ii. breastfeeding and nutrition, iii. Family planning iv. Postnatal care etc.

The discussion on spousal support this section could also be moved to the relevant section 

Lines 353-414: Thank you for this advice. The section on postnatal care and challenges related to gender roles has been reorganized accordingly. 

Discussion

This statement in line 361 needs to be re-written as it comes out as a strong statement likely just due to the wording “not a well-established practice for all mothers in Buikwe district.”

Line 278: The sentence has been re-written. 

This comes out in the discussions but is not adequately discussed in the results “young women’s low knowledge about issues like 381 hygiene and how to care for newborns”

Line 267-274: The mentioned concern is now explained more extensively in the result section. 

“Challenges related to logistics and economy were explained as reasons for suboptimal experiences both from the user and provider perspective.”- This needs to be further explained as it is not clear and also a strong statement

Line 248: The sentence has been slightly re-written and further explained. 

The UGC comes up again in the discussion but is not well discussed in the background/ methods so that one can be able to relate to it in the discussion

Line 51: UCG is now more extensively mentioned in the background

Minor comments

Some terminologies that come out as a bit strong making some statements difficult to interpret

“In acquiring to understand” change wording here

Line 214: This sentence has been replaced

Line 132 “TBAs, 3 VHTs” write in full

Acronyms have been spelled out when first used and late as abbreviations. 

Discrepancies in the use of abbreviations throughout the document for example as above and in the results and discussion sections

Thank you for noticing and pointing it out. 

Reviewer #3: 

Overall comments:

This is an important piece of work that is addressing a key global health priority. However, the themes reported on maternal experiences of health care are very similar to what has been found in other contexts in sub-Saharan Africa, so it is ever so important that the authors draw out what their key messages are and what this adds to the literature. They also need to link it better to neonatal outcomes as they have stated in the title. Unfortunately, most sections are not written at the level required for a publishable manuscript. Some of the sections are very long and lack focus (e.g. Introduction and Results). The Methods section needs to be reorganised and, in some sections, explained in more depth. The whole manuscript would benefit from a review of the grammar.

Thank you for appreciating the importance of this topic and for your detailed and constructive review. The key-messages have been more clarified and the whole manuscript has been carefully revised and re-structured. The grammar has been revised. 

Title: Women’s access to and use of maternal health care and home support systems to

reduce neonatal deaths in Buikwe district, Uganda: a qualitative study

I appreciate the complexity of what was being explore in this study but suggest the title captures this work more succinctly. For example:

Women’s experiences of maternal and newborn health care services in Buikwe district, Uganda: a qualitative study

Thank you for clarifying the title to be more specific, the suggested title has replaced the former, with some adjustment. We included “and support systems”, as this has a big focus in the paper. 

New title: Women’s experiences of maternal and newborn health care services and support systems in Buikwe district, Uganda: a qualitative study.

Abstract

Background: This study aimed at describing maternal support systems – revise to This study aimed to describe….

This is now corrected

Methods and findings:

Data Management is missing. 

Line 30: Data management has been described. 

Line 43…prohibited women in receiving optimal – replace in with from. 

Has been corrected

Line 38 Unclear how “ Postnatal follow-ups were found unsatisfactory.” This needs to be explained a bit more in the Abstract as it is linked to a recommendation in the Conclusion.

Line 39: Additional clarifying information has been added.

Please review the grammar in the Abstract.

A review of the grammar has been conducted. 

Introduction

This section needs to be a lot more succinct and linked to the aims of the study. As written, it is too long. Some of this content can be moved to the Discussion.

The introduction section has been modified and shortened. 

Some specific points:

Line 47 … 2030 if today’s trend continues- revise to: if the current trend continues.

This point has been revised. 

Line 51-53…. For the mothers it is also important to have someone to trust and rely on during pregnancy, time of childbirth and the following post-partum period, and by such providing an optimal start in life for the newborns (3). So that…please expand on this.

Lines 67-70: Further elaboration has been implemented.

Methods

Line 108 …individual interviews – I am not clear what individual interviews are because in qualitative research, interviews are conducted between a researcher and a participant and not a group. Please clarify.

Sorry about the confusion. Key informant interviews is correct. 

Line 108…key-informants- do you mean key informant interviews?

Yes, key informant interviews. This is now altered.

Line 113- A summary of methods is given below: please delete this as it doesn’t provide any additional

This sentence is now deleted

Line 127- Participants were selected purposefully – based on what characteristics? Please expand. (mentioned in line 149-150)

Lines 143-147: Characteristics and a table have been included, both for key-informants and mothers. 

Line 129- Sample – This subtitle should be “Sampling”. It would be useful to have a Table linked to this section to summarise the sampling strategy.

A table describing characteristics of participants has been created and included (line 147), as well as a flow chart displaying an overview of the distribution of participants (line 135). 

Line 176 Instruments This should be incorporated at the beginning of the sub-section: Data Collection.

Lines 150-153:Instruments are now incorporated under Data collection

Data Management is missing.

Line 184: A paragraph about data management has been added. 

Results

Introductory paragraph needs to be revised as it doesn’t lead the reader well into the content of the results. This section is also long so the authors need to decide what the key messages are and re-write this in a logical manner that brings out the key messages.

The Introduction has been revised and shortened and more context and focus have been given to key points in the paper. 

Discussion

This section of the manuscript is more structured with clearer messages of the relevance of this work in the context of existing literature on the topic and context. However, the focus on maternal experiences doesn’t link adequately to neonatal outcomes as stated in the title so I wonder whether the paper should focus on maternal experiences of accessing health care.

The title and aim of the study have been altered to focus more on women’s experiences with the health care system as suggested. 

Conclusion

No comments

---

## [Decision Letter · Decision Letter 1]

22 Oct 2021

PONE-D-21-14606R1Women’s experiences of maternal and newborn health care services and support systems in Buikwe district, Uganda: a qualitative studyPLOS ONE

Dear Dr. Rød,

Thank you for submitting your manuscript to PLOS ONE. Peer review of your manuscript is now complete but as pointed out by the third reviewer, there are grammatical errors in all sections that need to be addressed. The reviewer has also commented on the length and focus of the Introduction section. Although the reviewer is concerned about the combined "Results and discussion" section, this format, though not popular, is acceptable by the Journal. Therefore, we invite you to submit a revised version of the manuscript that addresses the points raised by the third reviewer, especially the English grammar issue. For example, the first part of the opening sentence of the Introduction is not clear.

We look forward to receiving your revised manuscript.

Kind regards,

Calistus Wilunda, DrPH

Academic Editor

PLOS ONE

Journal Requirements:

Reviewers' comments:

Reviewer's Responses to Questions

**Comments to the Author**

1. If the authors have adequately addressed your comments raised in a previous round of review and you feel that this manuscript is now acceptable for publication, you may indicate that here to bypass the “Comments to the Author” section, enter your conflict of interest statement in the “Confidential to Editor” section, and submit your "Accept" recommendation.

Reviewer #1: All comments have been addressed

Reviewer #2: All comments have been addressed

Reviewer #3: (No Response)

2. Is the manuscript technically sound, and do the data support the conclusions?

Reviewer #1: Yes

Reviewer #2: Yes

Reviewer #3: Yes

3. Has the statistical analysis been performed appropriately and rigorously? 

Reviewer #1: Yes

Reviewer #2: N/A

Reviewer #3: N/A

4. Have the authors made all data underlying the findings in their manuscript fully available?

Reviewer #1: Yes

Reviewer #2: Yes

Reviewer #3: No

5. Is the manuscript presented in an intelligible fashion and written in standard English?

Reviewer #1: Yes

Reviewer #2: Yes

Reviewer #3: No

6. Review Comments to the Author

Reviewer #1: Thank you for inviting me to review this paper again and great job on your edits and addressing my previous comments and concerns. I enjoyed your paper and research.

Reviewer #2: (No Response)

Reviewer #3: Overall comments:

Thank you for addressing most of my comments. This manuscript reads much better but there are grammatical errors in all sections that need to be addressed, as it cannot be published like this. Other aspects that need to be addressed:

Introduction:

This remains very long and lacks focus. Please limit this to 3-4 paragraphs that really draw the reader to the aims/objectives of the study. Some of the content here can be moved to the Discussion.

Results

The sub-title is now “Results and discussion” which is incorrect. This should just be Results.

The current format that combines results and the discussion for a specific result is not one that I am familiar with. I think this should use the standard format of a Results section then Discussion.

The content in the Results and Discussion seems appropriate. I would like to see how the Discussion is written in one section and not combined with the Results.

7. PLOS authors have the option to publish the peer review history of their article (what does this mean?). If published, this will include your full peer review and any attached files.

Reviewer #1: No

Reviewer #2: No

Reviewer #3: No

---

## [Author Response · Author response to Decision Letter 1]

18 Nov 2021

PONE-D-21-14606R2

Women’s experiences of maternal and newborn health care services and support systems in Buikwe District, Uganda: A qualitative study

Dear Editor and Reviewers at PLOS ONE, 

The authors appreciate the constructive and thorough reviews. We have adhered to the suggestion of the third reviewer and the manuscript is now in standard format with Discussion following Results. The manuscript has been professionally proofread to improve the grammar. A point-by-point response letter to the comments suggested by the editors and reviewer is given below. As requested, we have updated the information in the Data Availability statement to comply with PLOS ONE submission guidelines. 

Sincerely, on behalf of the authors, 

Marte Bodil Roed. 

PLOS ONE

Journal Requirements:

The Reference list has been updated and is complete. Some citations and references have been omitted due to editing and reduced length of the paper. 

A) Please provide non-author contact information for a data access committee, ethics committee, or other institutional body to which data requests may be sent.

The Data Availability statement now includes the name and e-mail address to the Centre for International Health, University of Bergen, Norway. Reasonable requests for data access may be sent to post@cih.uib.no.

Reviewers' comments:

(Peer review: PONE-D-21-14606_R1)

Thank you for addressing most of my comments. This manuscript reads much better but there are grammatical errors in all sections that need to be addressed, as it cannot be published like this. 

The manuscript has undergone academic proofreading by a professional company approved and affiliated by the University of Bergen (Semantix). Suggestions and changes have been discussed among the authors and implemented accordingly. 

Other aspects that need to be addressed:

Introduction:

This remains very long and lacks focus. Please limit this to 3-4 paragraphs that really draw the reader to the aims/objectives of the study. Some of the content here can be moved to the Discussion.

The first sentence in the abstract and introduction has been re-written.

The introduction section has been reduced to 4 paragraphs and several components have been moved to discussion or omitted. It should now point more clearly towards the objectives of the study. 

Results

The sub-title is now “Results and discussion” which is incorrect. This should just be Results.

The sub-titles Results and Discussion are now separated. 

The current format that combines results and the discussion for a specific result is not one that I am familiar with. I think this should use the standard format of a Results section then Discussion. 

The authors agree that the format used in this paper is unusual. We have considered the various formats and the Results and Discussion sections are now again written in standard format in separate sections. The discussion has been edited and elaborated with contents added from the Introduction section. 

The content in the Results and Discussion seems appropriate. I would like to see how the Discussion is written in one section and not combined with the Results.

Thank you for your clear views and frank suggestions. The Result and Discussion sections are now separated.

---

## [Editor Report · Decision Letter 2]

2 Dec 2021

Women's experiences of maternal and newborn health care services and support systems in Buikwe District, Uganda: A qualitative study

PONE-D-21-14606R2

Dear Dr. Rød,

We’re pleased to inform you that your manuscript has been judged scientifically suitable for publication and will be formally accepted for publication once it meets all outstanding technical requirements.

Kind regards,

Calistus Wilunda, DrPH

Academic Editor

PLOS ONE
---

## [Editor Report · Acceptance letter]

6 Dec 2021

PONE-D-21-14606R2 

Women’s experiences of maternal and newborn health care services and support systems in Buikwe District, Uganda: A qualitative study 

Dear Dr. Roed:

I'm pleased to inform you that your manuscript has been deemed suitable for publication in PLOS ONE. Congratulations! Your manuscript is now with our production department. 

Kind regards, 

on behalf of

Dr. Calistus Wilunda 

Academic Editor

PLOS ONE